# Therapeutic Efficacy of Pharmacological Ascorbate on Braf Inhibitor Resistant Melanoma Cells In Vitro and In Vivo

**DOI:** 10.3390/cells11071229

**Published:** 2022-04-05

**Authors:** Heike Niessner, Markus Burkard, Christian Leischner, Olga Renner, Sarah Plöger, Francisco Meraz-Torres, Matti Böcker, Constanze Hirn, Ulrich M. Lauer, Sascha Venturelli, Christian Busch, Tobias Sinnberg

**Affiliations:** 1Department of Nutritional Biochemistry, Institute of Nutritional Sciences, University of Hohenheim, Garbenstraße 30, 70599 Stuttgart, Germany; markus.burkard@uni-hohenheim.de (M.B.); christian.leischner@uni-hohenheim.de (C.L.); olga.renner@uni-hohenheim.de (O.R.); sascha.venturelli@uni-hohenheim.de (S.V.); 2Department of Dermatology, Division of Dermatooncology, University of Tuebingen, Liebermeisterstr 25, 72076 Tuebingen, Germany; sarah.ploeger@nmi.de (S.P.); francisco.meraz-torres@med.uni-tuebingen.de (F.M.-T.); matti-till.boecker@student.uni-tuebingen.de (M.B.); constanze.hirn@med.uni-tuebingen.de (C.H.); 3Cluster of Excellence iFIT (EXC 2180) “Image Guided and Functionally Instructed Tumor Therapies”, Eberhard Karls University of Tuebingen, 72076 Tuebingen, Germany; 4Department of Internal Medicine VIII, University Hospital Tuebingen, Otfried-Mueller-Strasse 10, 72076 Tuebingen, Germany; ulrich.lauer@uni-tuebingen.de; 5Department of Vegetative and Clinical Physiology, Institute of Physiology, University of Tuebingen, Wilhelmstraße 56, 72074 Tuebingen, Germany; 6Dermatologie zum Delfin, Stadthausstrasse 12, 8400 Winterthur, Switzerland; dermadelfin@hin.ch; 7Department of Dermatology, Venereology and Allergology, Charité-Universitätsmedizin Berlin, Charitéplatz 1, 10117 Berlin, Germany

**Keywords:** malignant melanoma, cancer, ascorbate, vitamin C, BRAF inhibitor, BRAF^V600E^, vemurafenib, D4M.3A melanoma cells

## Abstract

High-dose ascorbate paradoxically acts as a pro-oxidant causing the formation of hydrogen peroxide in an oxygen dependent manner. Tumor cells (in particular melanoma cells) show an increased vulnerability to ascorbate induced reactive oxygen species (ROS). Therefore, high-dose ascorbate is a promising pharmacological approach to treating refractory melanomas, e.g., with secondary resistance to targeted BRAF inhibitor therapy. BRAF mutated melanoma cells were treated with ascorbate alone or in combination with the BRAF inhibitor vemurafenib. Viability, cell cycle, ROS production, and the protein levels of phospho-ERK1/2, GLUT-1 and HIF-1α were analyzed. To investigate the treatment in vivo, C57BL/6NCrl mice were subcutaneously injected with D4M.3A (Braf^V600E^) melanoma cells and treated with intraperitoneal injections of ascorbate with or without vemurafenib. BRAF mutated melanoma cell lines either sensitive or resistant to vemurafenib were susceptible to the induction of cell death by pharmacological ascorbate. Treatment of Braf^V600E^ melanoma bearing mice with ascorbate resulted in plasma levels in the pharmacologically active range and significantly improved the therapeutic effect of vemurafenib. We conclude that intravenous high-dose ascorbate will be beneficial for melanoma patients by interfering with the tumor’s energy metabolism and can be safely combined with standard melanoma therapies such as BRAF inhibitors without pharmacological interference.

## 1. Introduction

Cutaneous melanoma is one of the most common malignancies, accounting for nearly 80% of deaths derived from skin cancer in the Caucasian population [1]. Characterized by a worldwide annual increase of new cases from 287,723 in 2018 [2] to 324,635 in 2020 [3], melanoma remains the most lethal type of skin cancer. The worldwide cumulative epidemiologic data confirm the continuous rise of melanoma incidence during the last decades [4]. After many years of melanoma treatment with chemotherapeutics with little effect on overall survival (OS) [5,6], the implementation of targeted therapies and targeted immune therapies has significantly improved prognosis. The mitogen-activated protein kinase (MAPK) signal transduction cascade is physiologically stimulated upon the binding of growth factors to a receptor tyrosine kinase (RTK) on the cellular surface and the recruiting of a large network of serine-or tyrosine kinases via subsequent activation of the rat sarcoma (RAS), rapidly accelerated fibrosarcoma (RAF)**,** mitogen-activated protein kinase (MEK), extracellular signal-regulated kinase (ERK) pathway to execute cellular functions such as cell-cycle progression, de-differentiation, and anti-apoptotic events [7,8]. Most importantly, its inappropriate and constitutive activation provides a potent promitogenic stimulus that is very common for human cancers [7,8]. Oncogenic mutations in the v-Raf murine sarcoma viral oncogene homolog B (BRAF) gene are present in about 50% of melanomas and cause fast disease progression with especially poor prognosis [9,10]. The most frequent BRAF^V600E^-mutation results in RAS-independent monomeric signaling, increased kinase activity, and constitutive MEK and ERK activation [11]. Thereby, BRAF kinase inhibitors (BRAFi) induce stress-mediated apoptosis in BRAF^V600E^ melanoma cells [12]. Therefore, the first potent pharmacotherapy for the treatment of unresectable or metastatic melanoma, carrying the BRAF^V600E^ mutation, was vemurafenib, an orally available small molecule BRAFi inhibitor, approved by the US Food and Drug Administration (FDA) in 2011 [13]. Vemurafenib treated patients showed rapid response to therapy as well as prolonged overall and progression-free survival [9,14,15]. Nevertheless, the therapeutic effects are only temporary and are compromised by the emergence of drug resistance. Thus, 50% of treated melanoma patients develop disease progression within six to seven months after the initiation of vemurafenib therapy [9,15,16,17]. Due to the development of resistances and the growing incidence, there is still an urgent need for additional therapy options for melanoma patients. The increasing incidence of melanoma cases also translates into high direct treatment costs as well as indirect costs associated with the loss of potential life-years and productivity [4].

L-ascorbic acid or vitamin C (ascorbate), an essential micronutrient for humans, has been suggested for years to play a role in cancer therapy and to improve the survival of patients with advanced cancers [18]. Ascorbate has a great reducing potential and reacts with many reactive oxygen and nitrogen species in vitro, modulating the negative effects of ROS. Thus, it protects cells from oxidative stress at submillimolar concentrations [19]. Vice versa, ascorbate proved to be cytotoxic to a number of cancer cell types with high sensitivity in comparison to benign cells at high concentrations [20,21,22]. Importantly, a series of in vivo studies demonstrated the antitumor efficacy of parenteral high-dose ascorbate [20,23,24,25]. Besides its promising anticancer effects, ascorbate was also well tolerated and appeared to decrease the side effects of chemotherapy [24] and even enhanced the chemosensitivity of ovarian cancer [26]. In line, several publications have investigated the effects of vitamin C on melanoma cells in the last four decades. It was already shown in 1980 that vitamin C is preferentially cytotoxic to melanoma cells [27] and growth inhibition was also confirmed in murine melanoma cells [28]. The prooxidative effects of high-dose vitamin C mediated apoptosis-like cell death in a caspase 8 independent manner In B16 melanoma cells [29].

In this study, we analyzed tumor cell killing by ascorbate in melanoma cells with a focus on BRAF mutated melanoma cells and the interaction with the BRAFi vemurafenib. Moreover, ascorbate efficacy was also assessed in an immunocompetent, isogenic, BRAF^V600E^-mutated C57BL6/N^Crl^ melanoma mouse model. Therefore, mice were subcutaneously injected with BRAF^V600E^-mutated D4M.3A melanoma cells [30] and treated with ascorbate, vemurafenib or the combination thereof.

## 2. Materials and Methods

### 2.1. Isolation and Culture of Human Cells

Experiments were performed in accordance with the Declaration of Helsinki. Melanoma cell lines (451Lu and Mel1617) were kindly provided by M. Herlyn (Wistar Institute, Philadelphia, PA, USA) or purchased from the ATCC (A375, SKMel19 and SKMel28). The murine cell line D4M.3A was kindly provided by Prof. Dr. Constance Brinkerhoff (Norris Cotton Cancer Center, Hanover, NH, USA). Mycoplasma infection in the cells was regularly checked using a Venor GeM Classic Mycoplasma Detection Kit (Minerva Biolabs). BRAFi resistant cells were generated using vemurafenib starting at 0.2 µM and increasing the concentration with every passage up to 2 µM respectively [31]. Double resistant cells were produced likewise with additional increasing concentrations of trametinib (up to 50 nM).

### 2.2. Signaling Pathway Inhibitors and Treatments

Vemurafenib was purchased from Selleck Chemicals, while ascorbate (Pascorbin^®^) was purchased from Pascoe pharmazeutische Praeparate GmbH.

### 2.3. Viability Assay

The viability of melanoma cells was assessed using the 4-methylumbelliferyl heptanoate (MUH) assay. Briefly, 2.5 × 10^3^ cells were seeded into cavities of a 96-well plate. After 24 h, cells were treated in hexaplicates for 72 h with increasing concentrations of vemurafenib (up to 20 μM) or ascorbate (up to 16 mM) either as a single or combinational treatment. Before the analysis, cells were washed with PBS, subsequently incubated with 100 μg/mL 4-methylumbelliferyl heptanoate diluted in PBS (1:100 dilution of 10 mg/mL stock) for 1 h at 37 °C. In viable cells, 4-methylumbelliferyl heptanoate is hydrolyzed by intracellular esterases and lipases producing the highly fluorescent 4-methylumbelliferone. The arising fluorescence (λ_ex_ 355 nm, λ_em_ 460 nm) was detected with a Tristar fluorescence microplate reader (Berthold Technologies). The intensity of fluorescence is indicative of the number of viable cells and based on that, the relative viable cell number remaining after treatment was calculated relative to the respective controls.

### 2.4. Cell Cycle Analysis

For cell cycle analysis, 2.5 × 105 cells were seeded into cavities of a 6-well plate and incubated for 2 h. Subsequently, the cells were treated with indicated concentrations and combinations of vemurafenib and ascorbate for 3 days. Treatment was carried out in trip-licates and DMSO (0.02%) was used as a solvent control. To analyze the cell cycle distribution, floating and adherent cells were harvested, permeabilized with 70% ice-cold eth-anol overnight, washed with PBS twice and resuspended in PBS with 50 μg/mL propidi-um iodide (Sigma-Aldrich) and 100 μg/mL RNAse A (AppliChem). After staining for 30 min in the dark, the distribution of the cells in the different cell cycle phases was detected with a BDTMLSR II flow cytometer (BD Biosciences) using the FACSDivaTM software (BD Biosciences).

### 2.5. ROS Reporter Assay

Lentiviral particles were brought in HEK 293T cells (Biocat, Heidelberg, Germany) via the lentiviral vector pEIGW roGFP2-Orp1 (#64993, addgene) and a second-generation packaging system (pMD2.G and psPAX2). To determine the reactive oxygen species (ROS) H_2_O_2_, lentiviral particles were used to transduce SkMel28 melanoma cells for the expression of H_2_O_2_ biosensor roGFP2-Orp1 [32]. Briefly, 5 × 10^4^ SkMel28 cells per well were seeded onto a 24-well plate for 5 h. SkMel28 melanoma cells were transduced in the presence of 8 μg/mL polybrene using a multiplicity of infection (MOI) of 5. GFP positive cells were isolated using FACS sorting technology (Aria I, BD Biosciences). H_2_O_2_ levels in melanoma cells (SkMel28) were measured by flow cytometry (LSR II, BD Biosciences) through the fluorescence ratio Pacific Orange 405 nM/Alexa Fluor 488 nM.

### 2.6. Western Blot

After 1 to 24 h incubation with the drugs, cells were lysed directly in the dish for 30 min on ice with buffer containing 10 mmol/L Tris pH 7.5, 0.5% Triton X-100, 5 mmol/L EDTA, 0.1 mmol/L phenylmethylsulphonyl fluoride, 10 mmol/L pepstatin A, 10 mmol/L leupeptin, 25 mmol/L aprotinin, 20 mmol/L sodium fluoride, 1 mmol/L pyrophosphate, 1 mmol/L orthovanadate. Lysates were cleared by centrifugation at 13,000× *g* for 30 min and 15 to 60 mg protein was subjected to SDS-PAGE and transferred to polyvinylidene difluoride (PVDF) membranes. Proteins were detected with antibodies against HIF-1α (Biozol #IHC-IW-PA 1041), GLUT-1 (Abcam #ab115730), PCNA (Abcam #ab92552), SVC1 (Santa Cruz #sc-30113), SVC2 (Santa Cruz #sc-30114) and the Cell Signaling Technology primary antibodies (ERK #9102, pERKThr202/Tyr204 #4376, and β-actin #4970) and horseradish peroxidase conjugated secondary antibodies (Cell Signaling Technology), using Pierce ECL (Thermo Fisher Scientific) and an Amersham Imager 600 (GE Healthcare LifeSciences).

### 2.7. Colony Formation Assay

For the colony formation assay, 1.0 × 10^3^ cells were seeded into cavities of a 12-well plate and incubated overnight. Subsequently, the cells were treated with indicated concentrations and combinations of vemurafenib and ascorbate for 7 days. Treatment was carried out in triplicates and DMSO (0.02%) was used as a solvent control. To analyze the number of formed colonies, the cells were fixed in 4% formalin and stained with 3% crystal violet solution (Sigma-Aldrich; Taufkirchen, Germany) in 80% methanol for 2 h.

### 2.8. In Vivo Mouse Experiment

All animal experiments were approved and performed in compliance with both European Union and German law and approved by local authorities (Regierungspraesidium Tuebingen, HT3/16). C57BL/6NCrl mice were purchased at the age of 6 weeks from the Charles River Laboratory (Wiga, Germany) and were housed in the animal care facility at the University of Tuebingen. Female C57BL/6NCrl mice were used for the experiments. After a one-week acclimation period, C57BL/6NCrl mice received a subcutaneous (s.c.) right flank injection of 3 × 10^5^ D4M.3A melanoma cells suspended in 50 μL sterile phosphate buffered saline (PBS, Sigma, Germany). Once tumors were established (tumor volume 50–80 mm^3^), the animals were randomly assigned to four groups (14 in the control group and eight per treatment group) and treatment was initiated. Mice of all treatment groups were given intraperitoneal (i.p.) injections. The BRAFi vemurafenib (Zelboraf^®^), was injected at 30 mg/kg body weight in 100 μL PEG300 (Sigma Aldrich) solution, five times weekly. Ascorbate was given daily as an injection at 3 g/kg body weight (PASCORBIN^®^, Pascoe pharmazeutische Praeparate GmbH, Giessen, Germany). To inflict minimal pain in mice and reduce the number of i.p. injections to a minimum, mixed syringes (max. 700 μL total volume for i.p. injection using a 23G needle) were used for combination treatments. The control group received the equivalent volume of solvent (NaCl (Sigma-Aldrich)) as treatment for a maximum of total 20 i.p. injections. Mouse weights were determined every second day through the experiment. Tumors were measured every day (volume = (length × width × width/2)). To facilitate comparison between the treatment groups, all mice in the four cohorts were euthanized (killed by CO_2_) on the same day. Tissue specimens, body fluids, and tumors were excised, collected, and stored for further investigations.

### 2.9. Plasma Ascorbate Assay

Approximately 1 h after the last application, the treated animals were sacrificed and blood was collected by cardiac puncture. Plasma was isolated by centrifugation at 1.650× *g* for 10 min and immediately snap frozen for storage at 80 °C. A protocol derived from Ma et al. was developed to determine ascorbate concentration in plasma [33]. Briefly, 50 µL of the plasma samples were diluted with 200 µL of 1× Hanks balanced Salt Solution (HBSS) and determined against an HBSS-diluted ascorbate standard series in a fingerstick blood glucose meter (GlucoCheck Advance, dm-drogerie markt, Germany).

### 2.10. Immunohistochemistry of Mouse Tumors and Organs

For immunohistochemical analysis, mouse tumor tissue was fixed in 4% formalin, embedded in paraffin, and stained with H&E. Proteins were detected with antibodies against HIF-1-alpha (Biozol #IHC-IW-PA 1041), GLUT1 (Abcam #ab115730, PCNA (Abcam #ab92552) and the Cell Signaling Technology primary antibodies (ERK #9102, pERKThr202/Tyr204 #4376, and b-actin #4970). Bound antibodies were detected using Ul-traView Universal Alkaline Phosphatase Red Detection Kits from Ventana (Tucson, AZ, USA).

### 2.11. TUNEL Staining

The TUNEL staining was performed using the Click-iT™ Plus TUNEL Assay for In Situ Apoptosis Detection, Alexa Fluor™ 488 dye from Thermo Fisher Scientific (Karlsruhe, Germany) according to the manufacturers’ protocol.

### 2.12. Statistics

Data were statistically analyzed with GraphPad Prism version 8.4 (GraphPad Software, San Diego, CA, USA). For multiple group comparisons, one-way ANOVA with subsequent Tukey’s multiple comparisons tests was used for *p*-value calculation and significance determination. *p*-values < 0.05 were considered statistically significant (*: *p* ≤ 0.05; **: *p* ≤ 0.01; ***: *p* ≤ 0.001; ****: *p* ≤ 0.0001).

## 3. Results

### 3.1. Growth Inhibitory Effects of Ascorbate on Melanoma Cells

To investigate the general effect of ascorbate on melanoma, different melanoma cell lines were used and treated with increasing concentrations of ascorbate. For every cell line the EC_50_ was calculated as presented in Figure 1. The cell lines tested did mainly harbor the oncogenic BRAF^V600E^ mutation (shown in red), but the panel also included one cell line, which was wildtype for BRAF and NRAS (shown in grey), and NRAS^Q61R^ mutated cell lines (shown in light grey). The distribution of the EC_50_ values was quite broad, ranging from 0.19 to 6.9 mM ascorbate. For the further study the focus was laid on BRAF mutated cell lines to additionally address the question of whether cells with acquired resistance to the standard therapy with a BRAF inhibitor still responded to ascorbate treatment.

To test the effects of ascorbate on BRAFi-sensitive and resistant cell lines, the viability was measured in five different cell lines with their respective BRAFi (vemurafenib) sensitive and BRAFi resistant variants. The cell lines 451Lu S, 451Lu R, A375 S, A375 R, Mel1617 S, Mel1617 R, SkMel19 S, SkMel19 R, SkMel28 S, and SkMel28 R were used, carrying the BRAF^V600E^ mutation, respectively. The best responses upon ascorbate treatment could be observed in the cell lines A375 R (Figure 2B, red curve), SkMel19 S (Figure 2D, black curve), and SkMel28 R (Figure 2E, red curve). The concentration of 1 mM ascorbate induced almost 100% cell death in these cell lines, and the corresponding IC_50_ values were the lowest (0.38 mM, 0.39 mM, and 0.34 mM). The lowest treatment response was observed in the cell line Mel1617 R (Figure 2C, red curve). One hundred percent cell death was only detectable when the cells were treated with 8 mM ascorbate and the highest IC_50_ (3.40 mM) was calculated. The other cell lines showed 100% cell death at about 2 mM (451Lu S (Figure 2A, black curve)), A375 S (Figure 2B, black curve), Mel1617 S (Figure 2C, black curve), and SkMel19 R (Figure 2D, red curve) or 4 mM (451Lu R (Figure 2A, red curve)) and SkMel28 S (Figure 2E, black curve) of ascorbate. IC_50_ values ranged from 0.59 mM to 1.6 mM. Different effects could be observed when comparing the IC_50_ values of the sensitive cells to their resistant derivates. A375 R and SkMel28 R tended to be more sensitive to ascorbate treatment when compared with the BRAFi sensitive parental cells. Contrary, 451Lu S, Mel1617 S, and SkMel19 S cells seemed to be more sensitive to ascorbate treatment than their vemurafenib-resistant derivates. The highest difference was detected in Mel1617. There, the IC_50_ value of Mel1617 R was 3.5× higher than the IC_50_ of Mel1617 S. However, all melanoma cell lines could be completely killed in the pharmacologically achievable range between 1–10 mM ascorbate. In order to show the relevance for ascorbate treatment also for patients that developed resistance against BRAFi + MEKi, we treated three double resistant (DR) cell lines with increasing concentrations of ascorbate. The IC_50_ value of A375 DR was the lowest with 1.49 mM, Mel1617 DR showed an IC_50_ of 4.27 mM and SKMel28 DR showed the highest IC_50_ value with 5.13 mM.

### 3.2. Effects of Ascorbate and Vemurafenib on Viability, Cell Cycle Distribution, and ROS Production of Melanoma Cells

To further investigate if the effect of ascorbate could be enhanced by adding the standard therapy in the form of a BRAFi and also to exclude that the effect of the standard therapy was diminished by ascorbate, the BRAF-mutated cell line SKMel28 was treated with ascorbate, vemurafenib, or the combination thereof. Whereas the effect of vemurafenib alone was only marginal, the effect of the combination did not significantly differ from the effect of ascorbate alone in the viability assay (Figure 3A,C). The cell cycle analysis revealed that the effect of ascorbate and vemurafenib could not be enhanced but was also not significantly reduced after combining them. Approximately 40% of the cells could be found in the subG1 fraction of SKMel28 S cells and up to 15.5% of the SKMel28 R cells (Figure 3B,D).

In order to measure ROS production in melanoma, a H_2_O_2_-sensitive GFP2 biosensor (roGFP2-Orp1) was expressed in SkMel28 melanoma cells by lentiviral transduction. Orp1 mediates the near-quantitative oxidation of roGFP2 by H_2_O_2_, and the Orp1-roGFP2 redox relay effectively converts physiological H_2_O_2_ signals into measurable fluorescence signals in living cells [32]. To ensure that the transduced cells respond equally to treatment with ascorbate, vemurafenib, or their combination, viability after treatment was measured and cell cycle analysis was performed. We observed that the viability was highly reduced by ascorbate and the combination (Figure 3E). In the cell cycle analysis, the ascorbate treatment could achieve a subG1 cell fraction of up to 20.8% which could not be significantly altered after adding vemurafenib (Figure 3F).

The fluorescence intensities measured by flow cytometry revealed that the signal was enhanced after all treatments, especially after the treatment with 2 mM ascorbate and the combination of 2 mM ascorbate and 5 µM vemurafenib. Therefore, it can be assumed that the H_2_O_2_ levels are strongly increased by the treatment with ascorbate, as well as the combination; which in turn leads to the decrease of viability.

### 3.3. Effects of Ascorbate and Vemurafenib on Viability, Cell Cycle, and Protein Expression of the Mouse Melanoma Cell Line D4M.3A

In the further course of the project, the combination was also tested in vivo in a preclinical mouse model. At first, we tested whether the cytotoxic effects of ascorbate were reproducible in the murine cell line D4M.3A. We observed that cellular viability was highly reduced by ascorbate and the combination (Figure 4A). In the cell cycle analysis, ascorbate treatment caused a subG1 cell fraction of 35.7%, and vemurafenib led to 21.9% of cells in subG1. These effects were not reduced by combinational treatment (Figure 4B). To address changes in the protein expression, whole cell lysates were analyzed after treatment with ascorbate, vemurafenib and the combination of both (Figure 4C). Interestingly, vemurafenib led to a hyperactivation of phospho-Erk1/2. This is known for cells, which are rather resistant against the therapy with BRAFi. The levels of Hif-1α and Pcna did not change much dependent on the treatment. Interestingly, the expression of Svct2 seemed to be rather strong and not affected by the combination, whereas the signal of Svct1 was relatively weak and was further reduced by the combination treatment. Glut-1, which is a downstream target of Hif-1α, was affected the most. After treatment with ascorbate, its expression already decreased whereas after vemurafenib and combination treatment it was nearly completely gone. In order to find out if a long-term treatment shows additional effects on growth inhibition, especially after combinational treatment, a colony formation assay was performed. Seven days of treatment with vemurafenib reduced the tumor cell growth. Eight mM of ascorbate (applied for one hour on days 1 and 3 after seeding of cells) completely prevented the formation of colonies. Importantly, combinations of 2 mM of ascorbate with vemurafenib resulted in better growth inhibition compared to the mono-therapies, revealing enhanced effects of the combination.

### 3.4. Effects of Ascorbate and Vemurafenib In Vivo

To assess the treatment efficacy in vivo, a C57BL/6NCrl mouse model was used to subcutaneously inject the isogenic murine melanoma cell line D4M.3A. Once the tumors were established, the animals were randomized into four groups (14 in the control group and eight in each treatment group), and treatment was initiated. Mice of all treatment groups were given i.p. injections. Vemurafenib was injected at 30 mg/kg body weight in 100 μL PEG300 (Sigma Aldrich) five times weekly. Ascorbate was given daily as an injection at 3 g/kg body weight. For the combinational treatment, both substances were directly mixed before application. The control group received the equivalent volume of solvent (NaCl (Sigma-Aldrich)) as treatment. Mouse weights were determined every second day throughout the experiment. Tumors were measured every day (volume = (length × width × width/2)). To facilitate comparison between the treatment groups, all mice in the four cohorts were euthanized by CO_2_ the same day (Figure 5A).

The ascorbate levels of five animals per group were determined in the plasma approximately 1 h after the last medication. Figure 5B shows that the groups receiving ascorbate treatment had elevated levels of plasma ascorbate compared to the animals which received the control treatment or vemurafenib alone. The tumor volumes of each tumor were calculated after the animals were sacrificed. All tumors of animals which received ascorbate, vemurafenib, or their combination showed a decreased size compared to the control tumors. The tumors from the group which received the combinational treatment showed even a significant reduction in volume compared to the control tumors (Figure 5C).

To monitor tumor growth over time, the measurements of each day and tumor were taken and related to the starting day (day 0) with 100% tumor volume (shown in Figure 5D). BRAF inhibition with vemurafenib (blue) reduced the tumor growth compared to the sham-treated control mice (black). Likewise, the treatment with ascorbate (orange) significantly diminished tumor growth. The combination therapy of ascorbate and vemurafenib (green) resulted in reduced tumor growth compared to the control, ascorbate, and vemurafenib treated groups. To assess the health status of the animals, different organs (lung, liver, gut, kidney, and spleen) were histologically analyzed after the experiment (Figure 6A). No differences in the organ structure or other cellular changes could be observed between the different treatment groups. The weight of the animals showed no significant differences between the groups during the whole experiment (Figure 6B). Furthermore, the tumors of the four different treatment groups were H&E stained (Figure 6C) as well as stained for Pcna to determine proliferation, Hif-1α as a hypoxia marker and for the glucose and dehydroascorbate transporter Glut-1 (Figure 6D). In line with the Western Blot data performed with the cell line D4M.3A shown above, the tumor tissue also showed no differences in the expression of Hif-1α. Pcna seemed to be more highly expressed in the control tumors and in the tumors treated only with vemurafenib. Glut-1 was reduced in all treated tumors compared to the control tumors. To further explore the induction of apoptosis, a TUNEL staining was performed, and indeed the highest signal for positive, apoptotic cells was observed in the tumors treated with the combination.

Taken together, our study showed that ascorbate should be a potent and well-tolerated additional treatment option for patients afflicted with BRAF mutated melanoma. It does not negatively interfere with the standard (BRAFi) treatment and even enhanced its therapeutic efficacy.

## 4. Discussion

To the best of our knowledge, we show for the first time that ascorbate has a cytotoxic effect on explicitly BRAF mutant melanoma cells including cells with acquired resistance to the BRAF inhibitor vemurafenib or BRAFi/MEKi double resistant cells. Similar data have been published by Yang et al. for the primary resistant melanoma cell line A2058 [34]. Ascorbate applied as Pascorbin^®^ in millimolar concentrations efficiently reduced viability in melanoma cell lines with BRAF^V600E^ mutations. Such concentrations were also measured in the mouse model used in this project one hour after intraperitoneal injection of vitamin C at 3 g/kg body weight. It should be noted that such blood levels can only be achieved by intravenous or parenteral administration of high doses of vitamin C, but not by oral intake [35]. In vitro cytotoxicity was accompanied by ROS induction as described by us and others [22,25,36]. In addition, we are the first to present data on the efficacy of the combined inhibition of the oncogenic driver BRAF by vemurafenib and high-dose ascorbate in BRAF-mutated melanoma cells. Interestingly, the combined therapy with vemurafenib plus ascorbate showed the most pronounced tumor growth inhibition in the D4M.3A melanoma mouse model when compared with the respective monotherapies. Tumor growth of D4M.3A was only moderately reduced by vemurafenib, in contrast to the literature [30], which may be attributed to differences in drug delivery. However, the cells also showed some primary resistance in our in vitro assays. Vitamin C has previously been shown to slow the tumor growth of B16 mouse melanomas in vivo [37]. Again, we observed a smaller effect, but ascorbate was administered only once daily in our study in contrast to the published B16 data. Our finding that ascorbate killed BRAFi-resistant and BRAFi/MEKi double resistant melanoma cells in a similar manner as the corresponding sensitive parental cells provides a plausible explanation for the enhanced combinatorial effect. However, the resistant cells used in this study might not represent the whole clinically observed repertoire of resistance mechanisms, as none of the resistant cell types used carried an additional NRAS or MEK mutation [31]. Therefore, we cannot conclude that high-dose vitamin C is effective against all resistant melanoma cell types, although this is very likely to be the case.

Similar results were found in other cancer entities with therapy resistant tumor cell subpopulations. In hepatocellular carcinoma (HCC), ascorbate was shown to be preferentially active against cancer cells with a stem-cell like phenotype (Sox-2, Oct-4, Lin28, or CD133 positive) in comparison to the cancer-stem cell marker negative liver cancer cells [38]. This cancer stem-like cell (CSC) phenotype was associated with an increased expression of sodium vitamin C transporter 2 (SVCT2) as major ascorbate transporter and thereby increased intracellular ROS and ATP depletion upon ascorbate treatment. Similar mechanisms might account for melanoma cells that also express such CSC markers [39,40,41]. Interestingly, A375 melanoma cells that escaped vemurafenib treatment showed a higher ability to form melanospheres [42], an accepted marker of cancer stem-like cells.

Although not relevant in the D4M.3A melanoma cells used here, HIF-1α was shown to promote stem-like behavior [43] and to be an inducer of the stem cell marker Nanog in B16 melanoma cells [44]. Proline hydroxylases (PHDs) target the protein for degradation in an ascorbate dependent manner. Therefore, ascorbate has the potential to downregulate HIF-1α and thereby reduce the self-renewal capacity and tumorigenicity of cancer cells.

In hematopoietic stem cells and blood cancer cells, pharmacological doses of ascorbate restored TET function while slowing leukemia progression. Ascorbate acted as an important cofactor for TET and other dioxygenases [45]. Pharmacological ascorbate might therefore represent a promising adjuvant to standard leukemia therapies. Although not directly applicable to solid cancers, loss or decreased TET activity with low 5-mC levels was also shown to be an epigenetic hallmark of malignant melanoma [46], and ascorbate treatment increased 5-mC levels in melanoma cells [47].

One general resistance mechanism to BRAF inhibition is the reactivation of the MAPK pathway. This can be achieved by a variety of events, including novel RAS and MEK mutations or epigenetic activation of alternative signaling proteins such as COT or various receptor tyrosine kinases [48,49,50,51], all of which lead to hyperactivation of the pathway and massive phosphorylation of ERK1/2. Su et al. demonstrated that ascorbate eradicated thyroid cancer cells by simultaneous inhibition of the MAPK/ERK and phosphatidylinositol 3-kinases (PI3K)/Proteinkinase B (AKT) pathways [52]. Interestingly, ascorbate selectively killed KRAS and BRAF mutant colorectal cells because of their high expression levels of GLUT-1 [53]. Mechanistically, high-dose ascorbate inhibits glycolysis via inactivation of glyceraldehyde 3-phosphate dehydrogenase due to intracellular high ROS levels. Although BRAF inhibition in melanoma cells is partially acting via down-regulation of GLUT-1 and GLUT-3, this is totally reversed upon resistance development as it was shown in in vitro melanoma models and in vitro patient samples [54]. We observed similar mechanisms in the D4M.3A cells, which showed reduced levels of Glut-1 upon BRAF inhibition and were further reduced with the combination treatment. The use of BRAF and MEK inhibitors was demonstrated to be associated with a metabolic switch from glycolysis to oxidative phosphorylation (OXPHOS) [55,56]. This switch renders survivor melanoma cells extraordinarily sensitive to ROS as shown by Corazao-Rozas et al. and Wang et al., who showed an increased sensitivity to ROS stress induced by elesclomol or HDACi in vemurafenib resistant melanoma cells [57,58]. In this regard, the concept of targeting BRAFi surviving cells with an ROS inducing and/or a GLUT-1/GLUT-3 shuttled drug is an attractive approach to avoid or prolong resistance development. Furthermore, it was shown that vemurafenib itself acts as an ROS inducer in melanoma cells at high concentrations [59,60]. Consistent with these findings, the combination of vemurafenib and pharmacological ascorbate proved to be more effective in reducing viability and inducing apoptosis in vitro and delaying tumor growth in vivo.

However, the mechanistic aspects underlying this combination effect need to be investigated in more detail. The downregulation of the transporter Glut-1 and Svct1 are probably not the only factors that contribute to the reduced tumor growth and higher number of TUNEL positive cells.

Despite the failure of a clinical trial (which applied too low dosages of vitamin C, did not control the ascorbate blood levels and used ineffective additional drugs) to demonstrate a clinically meaningful activity with ascorbate-drug combinations in melanoma patients [61], we assume that combinations of ascorbate with effective targeted therapies addressing the MAPK pathway could make a difference. Although we did not observe any toxicity in the mouse model, long-term application in patients needs to be carefully surveilled and clinical trials are needed to investigate this interesting combination to improve the results of targeted therapy of BRAF mutated malignant melanoma.

This strengthens a concept of the clinical combination of BRAF inhibition and high-dose ascorbate to tackle melanoma cells with acquired resistance to the BRAF inhibition and to exploit potential additive effects of their combination.

## 5. Conclusions

Our data strengthen a concept for the clinical combination of BRAFi with high-dose ascorbate to target melanoma cells, particularly to counteract acquired resistance to BRAF inhibition, but also to exploit potential additive effects of their combination. Clinical trials are needed to further investigate the role of ROS-inducing and redox-active drugs in combination with MAPK pathway inhibition in melanoma patients.

## Figures and Tables

**Figure 1 cells-11-01229-f001:**
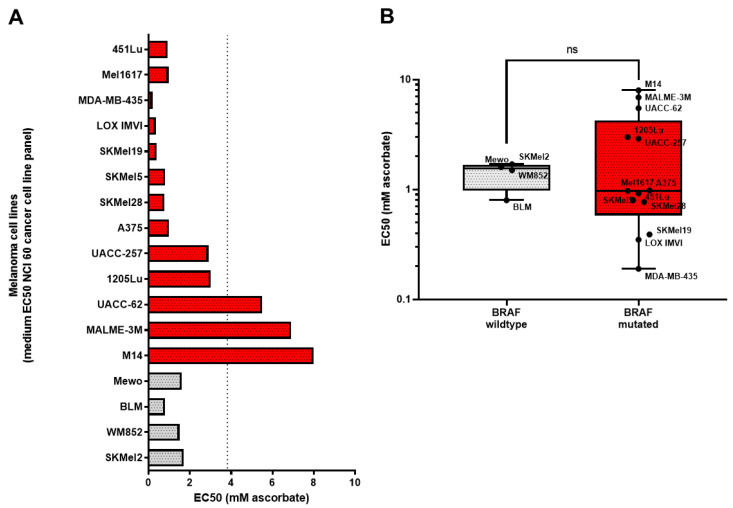
Growth inhibitory effects of ascorbate on melanoma cells. (**A**) EC50 of ascorbate treatment of melanoma cell lines including BRAF V600E mutated cell lines in red and BRAF wildtype cell lines in grey. (**B**) Median EC50 of ascorbate treatment from BRAF V600E mutated melanoma cells (red) was compared to the median EC50 of ascorbate treatment from BRAF wildtype melanoma cell lines (grey). *p*-value was calculated using the two-tailed Mann–Whitney test. ns: *p* > 0.05.

**Figure 2 cells-11-01229-f002:**
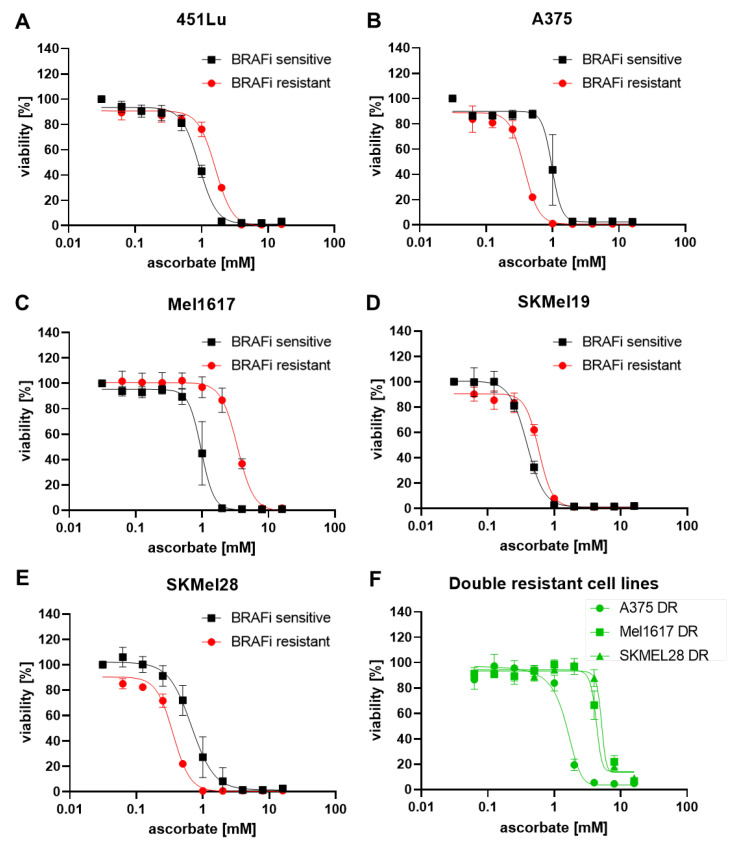
Growth inhibitory effects of ascorbate on melanoma cells.: (**A**–**E**) 451Lu (**A**), A375 (**B**), Mel1617 (**C**), SKMel19 (**D**), SKMel28 (**E**) and double-resistant/DR (**F**) melanoma cell lines were treated with different concentrations of ascorbate. The viability of treated cells was measured by the MUH assay and compared to the untreated control. The BRAFi sensitive cell lines are shown in black and the BRAFi resistant cell lines are shown in red. Shown are the mean values with standard deviations of three independent experiments, each measured in hexaplicates and normalized to the untreated control.

**Figure 3 cells-11-01229-f003:**
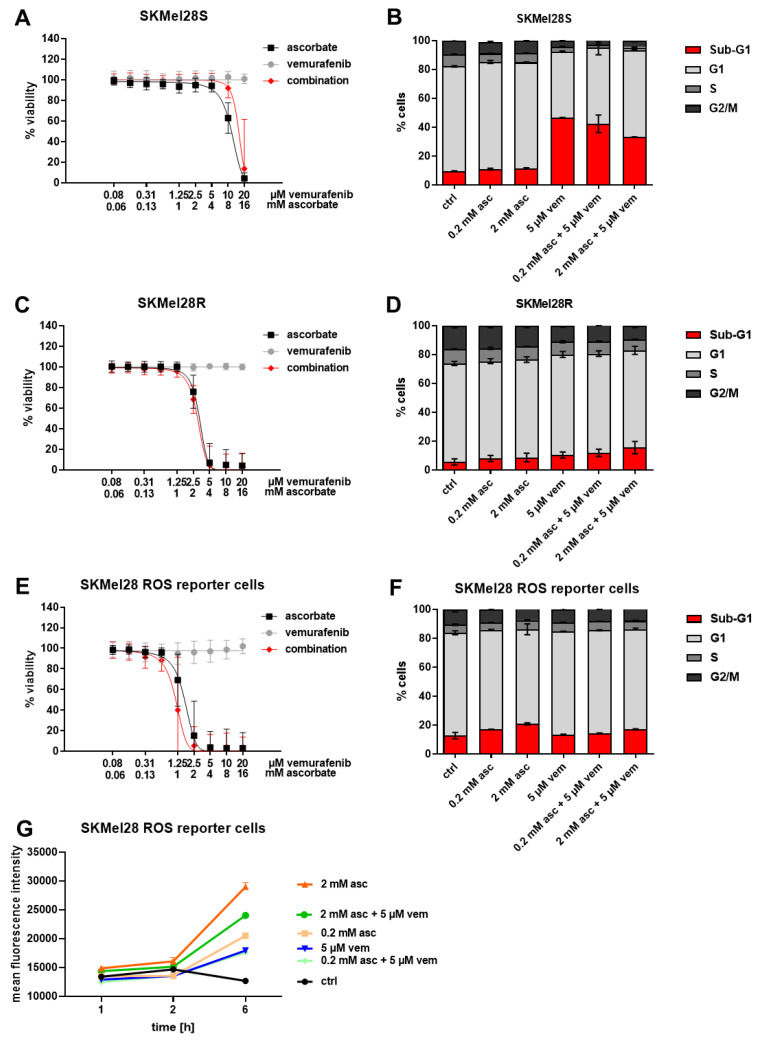
Effects of ascorbate and vemurafenib on viability, cell cycle distribution, and ROS production of melanoma cells. (**A**) SKMel28 S, (**C**) SKMel28 R, and (**E**) SKMel28 ROS reporter cells were treated with ascorbate, vemurafenib, and their combination. Depicted is the viability of treated cells compared to the untreated control. Shown are the mean values with standard deviations of three independent experiments, each measured in quadruplicates and normalized to the untreated control. (**B**) SKMel28 S, (**D**) SKMel28 R and (**F**) SKMel28 ROS reporter cells were treated with ascorbate, vemurafenib, and their combination. The cell cycle after PI staining was analyzed by flow cytometry. Shown are the mean values of each cell cycle fraction with standard deviations of three independent experiments, each measured in triplicates. (**G**) Flow cytometry measurement of the ratio Pacific Orange 405 nM/Alexa Fluor 488 nM, which resembles the amount of produced H_2_O_2_. SKMel28 H_2_O_2_ reporter cells were treated with ascorbate, vemurafenib, and their combination. Shown are the mean values with standard deviations of three independent experiments, each measured in quadruplicates and normalized to the untreated control.

**Figure 4 cells-11-01229-f004:**
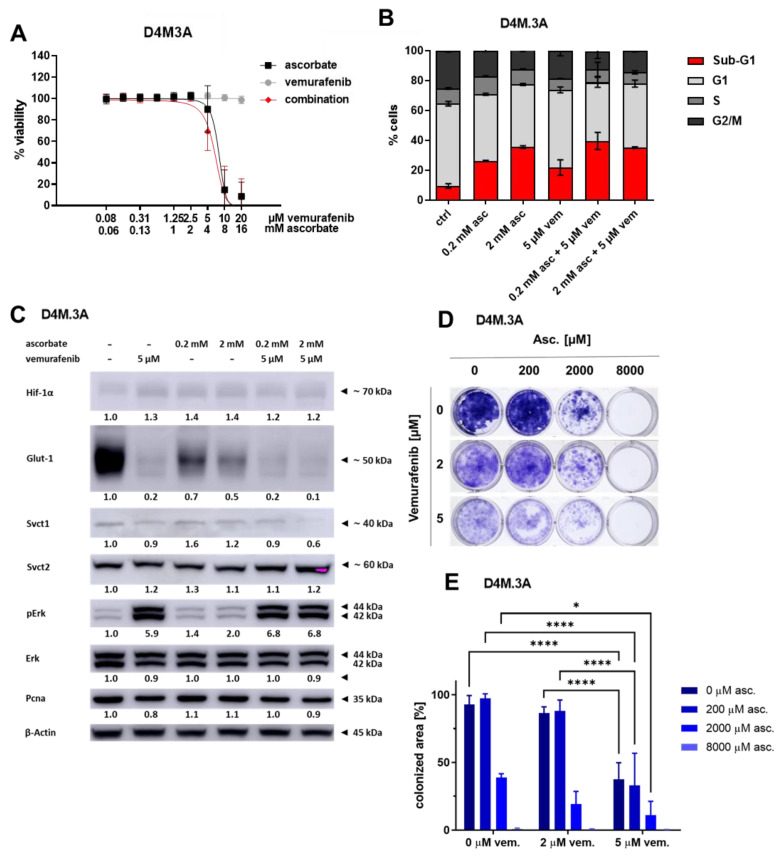
Effects of ascorbate and vemurafenib on viability, cell cycle, protein expression and long-term growth of BRAF mutated mouse melanoma cells D4M.3A. (**A**) D4M.3A cells were treated with ascorbate, vemurafenib, and their combination. Viability of treated cells was measured and compared to the untreated control. Shown are the mean values with standard deviations of three independent experiments, each measured in quadruplicates and normalized to the untreated control. (**B**) D4M.3A cells were treated with ascorbate, vemurafenib, and their combination. After propidium iodide staining, the cell cycle was measured by flow cytometry. Shown are the mean values of each cell cycle fraction with standard deviations of three independent experiments, each measured in triplicates. (**C**) Protein expression levels of Hif-1α, Glut-1, Svct1, Svct2, phospho-Erk1/2 (pErk), Erk1/2 (Erk), Pcna, and β-Actin as control were analyzed by Western Blot in the cell line D4M.3A after treatment with ascorbate, vemurafenib, and their combination. Shown is one representative experiment of three. (**D**) For a colony formation assay D4M.3A cells were treated with ascorbate, vemurafenib and the combination. The growth area of the formed colonies was measured and normalized to the surface area of the culture dish (100%). (**E**) Quantification of the colony formation assay shown in (**D**). D4M.3A cells were treated with ascorbate, vemurafenib and the combination. *: *p* ≤ 0.05; ****: *p* ≤ 0.0001.

**Figure 5 cells-11-01229-f005:**
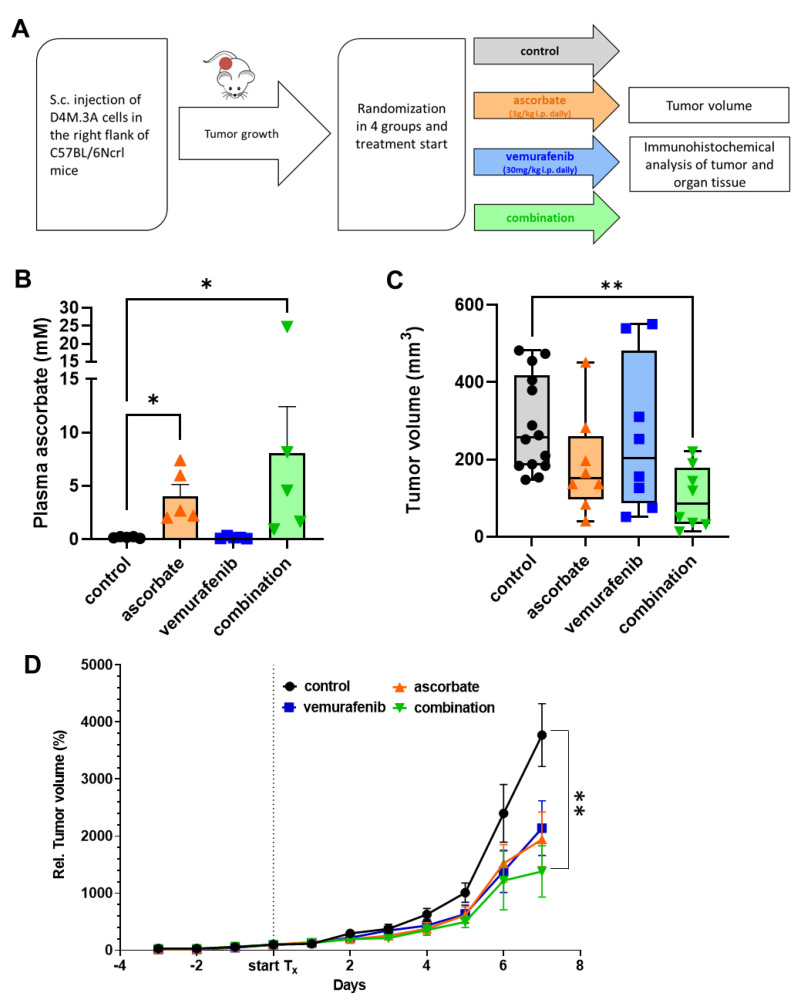
Effects of ascorbate and vemurafenib in an in vivo C57BL/6NCrl mouse model with subcutaneously injected isogenic D4M.3A melanoma cells. (**A**) Treatment scheme of in vivo mouse experiment. (**B**) Measurement of blood plasma levels of ascorbate in solvent injected control mice and mice receiving ascorbate (3 g/kg i.p., daily), vemurafenib (30 mg/kg i.p., five times weekly), and their combination; (**C**) Endpoint tumor volume measurement after 7 days of treatment with ascorbate (3 g/kg i.p. daily), vemurafenib (30 mg/kg i.p., five times weekly), and their combination; (**D**) Measurement of the tumor volume during treatment with ascorbate (3 g/kg i.p., daily), vemurafenib (30 mg/kg i.p., five times weekly), and their combination. The tumor volume was normalized to the tumor volume at the start of treatment to get the relative tumor volume. Shown are the mean values of normalized tumor volume +/− SEM. *: *p* ≤ 0.05; **: *p* ≤ 0.01.

**Figure 6 cells-11-01229-f006:**
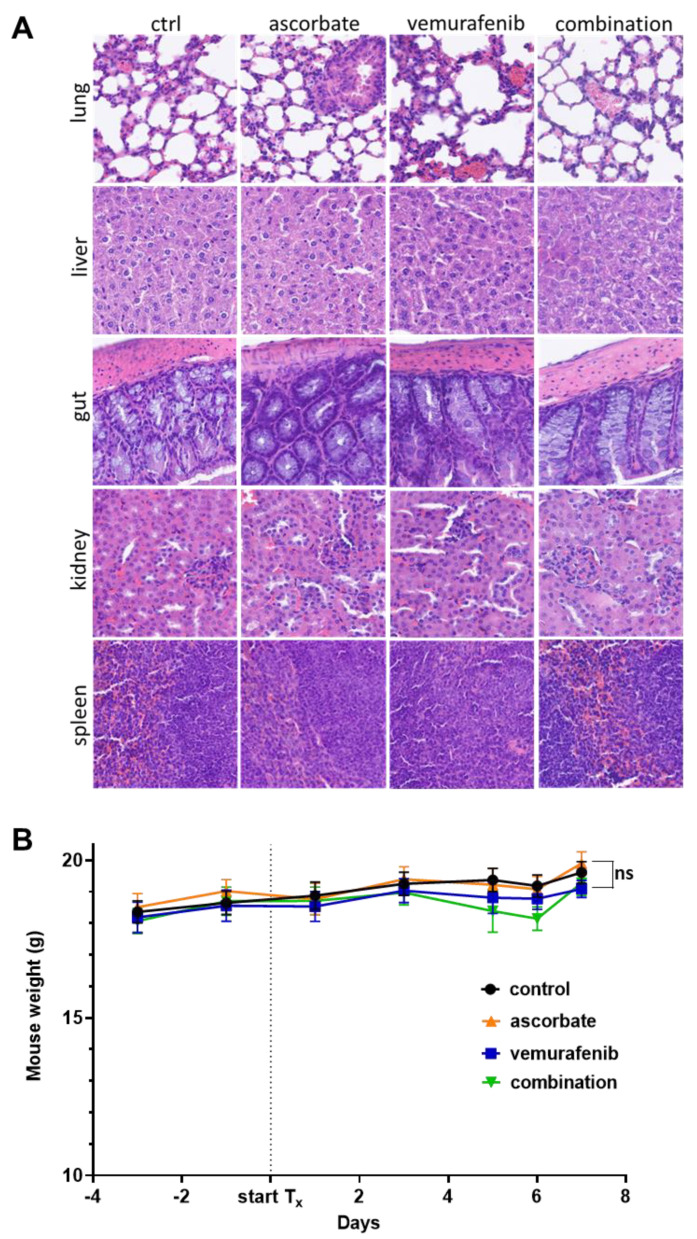
Treatment effects on mouse organs and weight and immunohistochemical analysis of mouse tumors. (**A**) H&E staining of mouse organs (lung, liver, gut, kidney, and spleen) of mice treated with ascorbate (3 g/kg i.p., daily), vemurafenib (30 mg/kg, five times weekly) and their combination. (**B**) Weight progress curves of mice during treatment with ascorbate (3 g/kg i.p. daily), vemurafenib (30 mg/kg i.p., five times weekly), and their combination. (**C**) H&E staining of tumors from mice treated with ascorbate (3 g/kg i.p., daily), vemurafenib (30 mg/kg i.p., five times weekly), and their combination. (**D**) Immunohistochemical staining for Pcna, Hif-1α, and Glut-1 of tumors from mice treated with ascorbate (3 g/kg i.p., daily), vemurafenib (30 mg/kg i.p., five times weekly), and their combination. (**E**) Immunofluorescent TUNEL staining of tumors from mice treated with ascorbate (3 g/kg i.p. daily), vemurafenib (30 mg/kg i.p., five times weekly), and their combination. Representative pictures are shown in a magnification of 200-fold. ns: *p* > 0.05.

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
