# Peer review of "Therapeutic Efficacy of Pharmacological Ascorbate on Braf Inhibitor Resistant Melanoma Cells In Vitro and In Vivo"

_cells, 2022, doi:10.3390/cells11071229_

Round 1
Reviewer 1 Report
This paper shows for the first time that ascorbate has a cytotoxic effect on BRAF mutant melanoma cells. The authors also are the first to present data on the efficacy of combined inhibition of the oncogenic driver BRAF 440 by vemurafenib and high-dose ascorbate in BRAF-mutated melanoma cells. I think the paper will be eligible to be published after minor revisions:
line 83you could add: "Ascorbate has a great reducing potential and reacts with many reactive oxygen and nitrogen species in vitro modulating negative effects of ROS; moreover Vitamin C protects cells from oxidative stress." and cite an article such as: doi: 10.1007/s13668-020-00322-4.
Good Luck!
Author Response
We thank the reviewer for his comment and followed his proposal to add the sentence: "Ascorbate has a great reducing potential and reacts with many reactive oxygen and nitrogen species in vitro modulating negative effects of ROS; moreover ascorbate in submillimolar concentrations protects cells from oxidative stress." in line 83 and also cited the article: doi: 10.1007/s13668-020-00322-4 at this point.
Reviewer 2 Report
This is an interesting study that tried to analyze therapeutic efficacy of ascorbate in melanoma (BRAFV600E) cell lines and cells with acquired resistance to BRAFi vemurafenib. Effects of ascorbate on D4M.3A melanoma tumor progression was tested using immunocompetent xenograft mouse model.
Although this is an interesting study, my enthusiasm is quenched by these major concerns:
- Did the authors consider including other BRAF inhibitors such as dabrafenib or encorafenib in addition to vemurafenib? What about comparison of ascorbate+vemurafenib combination to a clinically used BRAFi+MEK inhibitors? There are much better models of BRAFi resistance commercially available such as isogenic cell lines (vemurafenib sensitive A375 BRAFV600E and vemurafenib resistant A375 BRAFV600E/NRASQ61K cell lines with clearly defined mutational profiles) rather than cells grown in media supplemented with BRAi with unknown changes as a result of acquired resistance.
- Do the authors have information on mutational profile of melanoma cells resistant to vemurafenib that they created for their experiments? There might be much more random mutations present.
- What is the mechanism? Authors should provide at least some mechanistic insights.
- Why was SKMEL28 cell line chosen for cell cycle and ROS production studies? Can authors provide ROS phenotypic studies for the murine melanoma cells used for in their vivo studies?
- 3A (BRAFV600E) cells were chosen for in vivo experiments in order to study the responses of melanoma to ascorbate in immunocompetent model. I wonder whether the authors considered using BRAFi (vemurafenib) resistant cells for in vivo experiments rather than vemurafenib sensitive cells. Using BRAFi resistant cells as a model would be more beneficial for melanoma patients who develop resistance to BRAFi treatment.
- Quantifications of immunoblot analysis are missing. Based on the data provided, there doesn’t seem to be any additional effect of ascorbate on levels of Glut-1 compared to vemurafenib only group. The same applies to ERK signaling.
- What about other ascorbate transporters such as sodium-dependent vitamin C transporter 1 and 2? Investigating the expression and regulation of SVCTs might shine some light on mechanisms of ascorbate on melanoma tumor formation, progression, and response to therapy.
- The length of the in vivo tumor growth study seems to be very short for being able to draw any conclusions as the size of majority of tumors is relatively very small (Fig. 5C, mean at ~ 200 mm3). I don’t see statistically significant change between the sizes of tumors between ascorbate only, vemurafenib only and combination treatment group. Based on the Fig. 5C, vemurafenib doesn’t affect the tumor growth what is inconsistent with previously published studies and these results don’t support the in vitro studies the authors provide. Significance levels are missing in Fig. 5D.
- There are already several studies on effects of ascorbate in melanoma which have to be referenced in introduction and discussed in the discussion section of this manuscript.
Author Response
We thank the reviewer very much for the constructive and useful comments. We think the additional references which are now integrated in the manuscript as well as the newly performed experiments and analyses significantly enhance the paper. In the following, we will discuss each of the reviewer's comments in detail.
1.
Thank you very much for that important point. We agree that it would be very nice to test the other clinically applied BRAF and MEK inhibitors like dabrafenib, encorafenib, cobimetinib, trametinib and binimetinib. However, this is beyond the scope of the actual project, which we consider as proof of concept study that BRAF inhibition can be combined with high-dose vitamin C treatment. We had chosen vemurafenib because it is well known to be associated with ROS formation (Bauer et al. 2017 https://pubmed.ncbi.nlm.nih.gov/28151482/ ) and resistant cells seem to have an Achilles heel with increased ROS levels (Wang, et al. 2018 https://pubmed.ncbi.nlm.nih.gov/29754815/ )
We also agree that there are a multitude of different resistance mechanisms including the evolvement of additional driver mutations such as NRAS or MEK mutations. However, the cell lines used within this study represent a valuable tool for studying resistance mechanisms and overcoming those and have been successfully published in several manuscripts (Beck et al. 2013 https://pubmed.ncbi.nlm.nih.gov/23362240/ , Sinnberg et al. 2016 https://pubmed.ncbi.nlm.nih.gov/27428425/ , Kosnopfel et al. 2017 https://www.ncbi.nlm.nih.gov/pmc/articles/PMC5482615/ , Makino et al. 2018 https://pubmed.ncbi.nlm.nih.gov/28415756/ ). In addition, we partially know the resistance mechanisms in these cell line and could show that they have a predominant activation of receptor tyrosine kinases and no mutation in NRAS codon 61 (Sinnberg et. al 2016). As mentioned above, we would highly appreciate such analyses with RAS or MEK mutated cells maybe even in patient-derived melanoma cells in the near future. We added to the discussion that we cannot conclude that vitamin C is effective for all kind of BRAFi resistant melanoma cells (page 17, line 458).
Although not directly requested by reviewer 2 but pointing in the same direction, we have added response data (viability data) of double resistant (against BRAFi and MEKi) cell lines (Kosnopfel et al. 2017) towards ascorbate in Figure 1F.
2.
The complete mutational profile of the resistant cells is not known. However, we have looked for additional NRAS or MEK mutations in a previous project by Sanger sequencing (Sinnberg et al. 2016 https://pubmed.ncbi.nlm.nih.gov/27428425/). As mentioned above, no such mutations were identified. We added this limitation to page 17, page 458.
3.
We agree that the mechanistic aspects are an important point for the acceptance of high-dose vitamin C as an effective anti-cancer drug. We believe that one of the most important and immediate early phenomena in vitro that does occur already after a 1 hour treatment is the development of ROS as previously published by us and others (Sinnberg et al. 2014 https://pubmed.ncbi.nlm.nih.gov/24330097/ , Chen et al. 2005 https://www.pnas.org/doi/10.1073/pnas.0506390102 ). We strongly believe that this is an important trigger of subsequent alterations like Hif1α reduction. Therefore, we used the ROS biosensor SKMel28 cells in Figure 3 to show this ROS induction within 6 hours. Additionally, we see in the mouse tumors that were treated with the combination an increased number of apoptotic cells (Figure 6E) and reduced Glut-1 expression, which might be an important reason for the slowed tumor growth. In vivo, the ROS induction occurs probably not in the whole tumor and therefore this metabolic reprogramming comes into play. In line with this, we saw by trend a reduced expression of the sodium dependent vitamin C transporter 1 (Svct1) with combined treatment (Figure 4C). We summarized the mechanistic results together with the obvious mechanistic limitations in the discussion “We observed similar mechanisms in the D4M.3A cells, which showed reduced levels of Glut-1 upon BRAF inhibition and were further reduced with the combination treatment.” (page 17, line 505ff).
We would like to add ROS data with the murine melanoma cells. But because a very short time frame for the revision this was not possible to do, because the cells need to be stable transfected with the ROS reporter plasmid to perform the assay analog to Figure 3G.
We have chosen the DM.3A cells because we wanted to use an immunocompetent mouse model for the combination experiment in vivo. Therefore, human BRAFi resistant melanoma cells were not applicable. Interestingly, we experienced that the 3A cells are rather resistant (primarily) to the vemurafenib inhibitor in vitro (Figure 4A) and required a longer exposure time as applied in the colony formation assay (new Figure 4D and 4E) to mediate clear effects. This explains also the effect of vemurafenib as mono-therapy in the in vivo experiment (slightly reduced tumor growth compared to the control).
Quantifications were added to the western blots in Figure 4C. We added the effect revealed by the semi-quantification to the text (see page 11, line 340 - 351).
We added SVCT1/2 western blots to Figure 4C. Interestingly, the expression of SVCT2 seemed to be rather strong and not affected by the combination, whereas the signal of SVCT1 was relatively weak and was further reduced by the combination treatment (Figure 4C).
Thanks for the totally correct hint. We agree that there is no significant change in the tumor volumes at the end of the treatments after day 7 for the mono-therapies with vemurafenib and ascorbate. In Figure 5C you can see only a trend of smaller tumor volumes. This is in line with the small effects of vemurafenib in vitro on the D4M.3A cells. As the reviewer might know, this cell type has a loss of Pten and might be therefore primarily (relative) resistant to BRAFi although they are described to be susceptible (what they partially are in our long-term exposure data).
We added the missing statistics to Figure 5D. We also discussed the effects of the mono- and combination therapies in terms of the literature in the revised discussion (page 17, line 460ff).
Thanks for the comment that helped improving the manuscript. We added the following original publications to the introduction whose authors were the first to describe the activity of vitamin C on melanoma cells:
Bram et al. Nature 1980 https://pubmed.ncbi.nlm.nih.gov/7366735/ , Gardiner and Duncan 1988, Kang et al. 2003 https://pubmed.ncbi.nlm.nih.gov/3149408/ https://pubmed.ncbi.nlm.nih.gov/12827307/ (page 2, line 91-96).

Reviewer 3 Report
The authors test the toxicity of a high dose of ascorbate acid (AA) in many melanoma cell lines. The EC50 ranges from 0.19 to 6.9mM. They also examine the killing effect of AA on BRAFi (vemurafenib) sensitive and resistant cell lines. The sensitivity of the cells to AA is different in different cell lines, which are not tightly bound with the cells that are sensitive or resistant to vemurafenib. Interestingly, all tested cell lines are completely killed in the range 1-10mM of AA which is pharmacological achievable. The subG1 fractions of cells are not significantly increased upon AA treatment. The combination treatment of vemurafenib and AA did not cause a significant increase in subG1 fraction when compared to vemurafenib-only treatment. AA treatment leads to the increase of ROS production. In the murine cell line D4M.3A, AA treatment caused a 35.7% of subG1 fraction, while vemurafenib result in 21.9% of subG1 fraction. The combination treatment did not further boost the effect. Finally, they tested the effect of AA in C57BL/6NCrl mouse strain with the isogenic line D4M.3A. The tumor volumes decrease significantly in the combined group.
Several major issues need to be considered before accepting the paper for publication.
- Many papers have been addressed AA has anticancer activity in melanoma, such as Nature 1980, 284: 629-631, Prostaglandins leukot essent fatty acids, 1988, 34(2):119-126, Cancer Immunol immunother 2003:693-698. The authors just list some reviews about AA in other cancers. It is strongly recommended the author review AA functions in melanoma.
- The authors did not review the signaling cascade of AA in cancers and melanoma. They did not compare the results of HIF-1 and Glu with other papers. It is not conclusive how AA works in anticancer activity by the signaling cascade provided in this paper.
- The authors did not review the clinical trial outcome in the paper. Some clinical outcomes are not promising in combination with other anticancer drugs, such as Melanoma Res 2008, 18(2):147-51. The author did include some reviews but fail to describe the results.
- Although AA treat melanoma is not new, the effects of AA in BRAFi (vemurafenib) resistant lines are not reported. The author should emphasize this in the abstract and the paper.
- The data of vemurafenib and AA treatment alone and in combination are different in their effects in vitro and in vivo. In vitro, the combination seems not to be effective, while in vivo the combination works well. The author should discuss this in the paper.
- Whether the monotherapy of AA and vemurafenib play a role in the animal model need to be discussed and compare the results with other paper.
- The authors need to discuss the concentration of AA reach in the blood after taking it orally which is more clinically relevant.
- It is suggested the author use normal cells to test the toxicity of AA at a high dose.
Author Response
We thank the reviewer very much for the constructive and useful comments. We think the additional references which are now integrated in the manuscript as well as the newly performed experiments and analyses significantly enhance the paper. In the following, we will discuss each of the reviewer's comments in detail.
1.
We thank the reviewer for the helpful comment. To follow the proposal, we added the following references (Bram et al. Nature 1980 https://pubmed.ncbi.nlm.nih.gov/7366735/ , Gardiner and Duncan 1988 https://pubmed.ncbi.nlm.nih.gov/3149408/, Kang et al. 2003 https://pubmed.ncbi.nlm.nih.gov/12827307/) to the introduction in order to present the current knowledge of AA and melanoma. (page 2, line 91-96).
2.
We apologize for the incomplete presentation of our data in the context of the literature. We tried to be more clear in the revised version also in terms of mechanism of AA on melanoma cells (or at least the melanoma cells presented here). 1st We showed that Glut-1 is reduced after treatment of the cells with either the BRAFi or AA but even stronger in the combination. The reduced capacity for glucose uptake might cause a reduced proliferation rate. 2nd Indeed we cannot detect changes in Hif1α levels in the cultured D4M.3A nor do we detect a strong Hif1α signal in the murine tumors. Therefore, we assume Hif1α not to play a dominant role in the reduced tumor growth and higher apoptotic rate.
3.
We thank the reviewer for that comment and added a short paragraph about trial outcome in the discussion where we also briefly discussed the results of such a trial. “Despite the failure of a clinical trial (which applied too low dosages of vitamin C, did not control the ascorbate blood levels and used ineffective additional drugs) to demonstrate a clinically meaningful activity with ascorbate-drug combinations in melanoma patients [61], we assume that combinations of ascorbate with effective targeted therapies addressing the MAPK pathway could make a difference.” The reference Bael et al., 2008 (https://pubmed.ncbi.nlm.nih.gov/18337652/ ) was also integrated [61]. (page 18, line 518ff.)
We thank the reviewer for strengthening the importance of our work with this hint. As recommended, we underlined the novelty of our work by adding the information about first in vitro data of combination of vitamin C and vemurafenib in melanoma cells with acquired resistance to MAPK pathway inhibitors. However, data about combination of BRAFi with vitamin C in primarily resistant melanoma cells do already exist (Yang et al. 2017 https://pubmed.ncbi.nlm.nih.gov/28370562/ ) and we added that information to the discussion as well (page 17, line445 ff).
We agree with the reviewer and try to resolve that issue with additional data. We already believed in the first version of the manuscript that a reason for the discrepancy between the in vitro and in vivo data is the duration and frequency of treatment. Whereas the cells in vitro were only treated for 1 hour and analyzed after 24 or 72 hours, the animals were treated daily with HDVC for one week. Therefore, we added a clonogenic growth or colony formation assay with treatment of vitamin C for 1 hour at days 1 and 3 and analysis after 7 days (in the presence or absence of the BRAFi vemurafenib) (Figures 4D and 4E). In line with the in vivo model, a clear combination effect was seen for the combination of 2 mM vitamin C with the BRAFi. Therefore, we believe that the most critical point that caused this difference is the long-term treatment in comparison with the short-term experiment.
6.
We added a discussion part about the effects of the mono-treatments with vemurafenib and AA to the manuscript and compared it to the literature (page 17, line 453ff).
7.
We thank the reviewer for the important point because it might not be clear to all readers. Oral administration cannot achieve the HDVC levels in the blood which were also detected in our mice after intra-peritoneal injection of ascorbate. This was clearly shown also for humans (Padayatty et. al 2004 https://pubmed.ncbi.nlm.nih.gov/15068981/). We added this point to the discussion (page 17, line 448ff).
8.
For a different project benign skin cells (fibroblasts) were treated with different concentrations of ascorbate. A significant reduction in viability could not be detected. We did not add the data to the manuscript, but we include them in the letter to the reviewer (see below).
Rebuttal Figure 1; CellTiter-Blue® Cell Viability Assay. The fluorescence is measured by 530Ex/590Em. Background fluorescence was subtracted from medium fluorescence and normalized to control (set as 100%). Additionally, as negative control, for cell death Triton X-100 (1% (v/v)) was used. The results are presented as means ± (SD) of three independent experiments made with three distinct donors. The viability of fibroblasts with 5x104 fibroblasts/well 24 hours after ascorbate treatment.
To further address the toxicity various organs of the mice were processed and examined and in neither of them a difference to the untreated control tissue could be detected (Figure 6A). The mouse weights during the treatment in vivo did not differ from the control treated mice and did not change significantly during 7 days of treatment, which is also a hint for HDVC not being toxic (Figure 6B).

Round 2
Reviewer 2 Report
Authors satisfactorily addressed most of my recommended changes.
Reviewer 3 Report
the author improves the paper significantly. it is ready for publish